# Position: Predicting AI's Impact on Labor Is a Core Machine Learning Problem

Yong Suk Lee [1]

## Abstract

Artificial intelligence is increasingly reshaping how work is performed, organized, and valued. Predicting AI's impact on labor is an interdisciplinary scientific question that examines how evolving AI capabilities interact with adoption, organizational change, and political and economic adjustments to reshape work and labor. This paper argues that predicting AI's impact on labor is a core problem for the AI and ML community to engage with, not solely a societal or ethical question, and that ML has a distinctive role to play in the broader interdisciplinary agenda. Questions at the center of modern AI and ML research, such as prediction under non-stationarity, distribution shift, endogenous feedback, and LLM-based agent simulation, are also core questions of economics and social science. This paper reviews current approaches in economics, management, and ML, identifies technical obstacles that limit existing prediction methods, and proposes a research agenda for AI- and ML-driven labor prediction.

## 1. Introduction

Artificial intelligence is reshaping how work is performed, organized, and valued (National Academies of Sciences, Engineering, and Medicine, 2025). AI systems already automate or augment a wide range of human tasks, often increasing productivity and reducing costs while also substituting for some workers (Hampole et al., 2025; Humlum & Vestergaard, 2025; Brynjolfsson et al., 2025b; Dell'Acqua et al., 2026). As AI capabilities advance, concerns have grown that automation may extend beyond individual tasks to entire roles or occupations (Frey & Osborne, 2017; Arntz et al., 2016). At the same time, AI's labor impact is more nuanced than a simple narrative of replacement (Autor et al., 2003; Acemoglu & Restrepo, 2019). Organizations adapt

their structures, workers adjust skill investments, and new roles emerge as technologies diffuse. Historically, technological change has reorganized work rather than eliminating it (Autor, 2015; Bresnahan et al., 2002; David, 1990), with effects that vary across occupations, firms, industries, and regions. Understanding AI's labor impact therefore requires grappling with dynamic adjustment, heterogeneity, and uncertainty.

More fundamentally, technology both shapes and is shaped by choice. What AI systems are designed to do reflects decisions made by developers, firms, and institutions. As such, AI's impact on labor is not external to the AI and machine learning (ML) community. In fact, the charter of OpenAI explicitly states that the goal of artificial general intelligence is to perform all economically valuable tasks (OpenAI, 2018). Given this stated objective, it is unsurprising that concerns about large-scale job displacement have become widespread (Cutter & Weber, 2025; Herrera, 2025; Roose, 2025; Herrera & Cutter, 2025).

This paper argues that predicting AI's impact on labor should be treated as a core machine learning problem. This builds on and goes beyond calls to consider labor impacts as an AI ethics, safety, or governance issue (Hazra et al., 2025; Acemoglu, 2024a; Gray & Suri, 2019; National Academies of Sciences, Engineering, and Medicine, 2025; International Labour Organization, 2021), but a scientific one. Predicting how evolving AI systems reshape work requires prediction under non-stationarity, distribution shift, endogenous feedback, and calibration under uncertainty, challenges that lie at the center of modern machine learning (Quinonero-Candela et al., 2009; Ovadia et al., 2019; Shmueli, 2010; Hartford et al., 2017; Perdomo et al., 2020; Chernozhukov et al., 2018; Athey & Imbens, 2019). Each are topics economics have examined for a long time under different names and angles: structural breaks and the Lucas critique (Lucas, 1976), Knightian uncertainty (Knight, 1921), external validity and extrapolation, endogeneity, simultaneity, and reverse causality.

Economics has developed tools for causal analysis and policy evaluation, as well as structural modeling of economic change. These methods are necessarily grounded in observed institutional and technological regimes. Recent research on AI and labor from the social sciences, in particular,

[1]University of Notre Dame, IN, USA. Correspondence to: Yong Suk Lee <yong.s.lee@nd.edu>.

*Proceedings of the 43rd International Conference on Machine Learning*, Seoul, South Korea. PMLR 306, 2026. Copyright 2026 by the author(s).

economics and management, has generated valuable insights into productivity, employment, task reallocation, and organizational change. However, predicting future change is harder when the task structure, occupations, and organizations themselves are shifting.

Predictive frameworks from ML can complement these methods by focusing on explicit targets, out-of-sample evaluation, distribution shift, and uncertainty. In addition, recent AI systems, particularly large language models, are capable of performing economically valuable work. As such, economic task performance has become an objective that shapes AI system design (Liang et al., 2022; Jimenez et al., 2024). LLM-based agents are also increasingly used as simulated economic actors, opening new ways to study labor market responses. Predicting how AI changes workflows, organizational structure, and labor demand is therefore closely aligned with the trajectory of ML research.

This paper is not arguing that any single model or framework can predict the complexity of the whole labor market or economy. Rather, the goal is to argue for interrelated prediction sciences built around various labor market targets (capability, adoption, workflow change, aggregate effects) that can be evaluated and updated over time. Predicting AI's labor impact is inherently interdisciplinary as it concerns technological capability, adoption, organizational change, economic impact, public opinion, and governance. Credible forecasting and evaluation infrastructure within the ML community, alongside LLM-based scientific advances, would help us better understand AI's labor market impacts (Park et al., 2023; Horton et al., 2026; Chopra et al., 2025).

The remainder of the paper proceeds as follows. Section 2 clarifies what it means to predict AI's impact on labor. Section 3 reviews current approaches and identifies key technical obstacles. Section 4 outlines research directions where ML methods can help overcome these obstacles. Section 5 discusses alternate views. We conclude with recommendations for how the ML community can advance this agenda.

## 2. What Does It Mean to Predict AI's Impact on Labor?

AI's impact on labor will manifest in many interrelated ways. Prediction in this context is not a single exercise, nor does it correspond to a single outcome. Rather, it involves anticipating changes at different levels of analysis and across multiple time horizons as AI capabilities evolve and diffuse through the economy. As AI systems are integrated into production, the tasks people perform may change (Autor et al., 2003; Acemoglu & Restrepo, 2019), how they allocate their time may shift, and their productivity may increase (Brynjolfsson et al., 2025b; Dell'Acqua et al., 2023; Peng et al., 2023; Noy & Zhang, 2023). Earnings, hours worked, and

job stability may change as well (Dominski & Lee, 2025), and these adjustments may be reflected in compensation and benefit structures. At the organizational level, firms may reorganize teams and workflows (Brynjolfsson et al., 2021; Lee et al., 2022). For example, functions such as marketing or human resources may require fewer workers as AI systems take on core responsibilities, while managers increasingly oversee AI agents alongside, or instead of, human employees. These changes are also accompanied by shifts in skill demand, shaping education decisions and interacting with labor supply constraints. As these adjustments spread across firms, labor demand may shift across occupations and skill sets (Alekseeva et al., 2021; Green, 2024). Even within the same occupational category, workers who are better able to use AI to create value may be rewarded differently, affecting wage dispersion and career trajectories (Acemoglu & Restrepo, 2022).

At the same time, the technology itself continues to evolve. Large language models, initially developed for next-token prediction, are now deployed to perform a wide range of economically relevant functions (Eloundou et al., 2023). Because a substantial share of modern work involves text, early applications disproportionately affected text-based occupations. As AI systems become increasingly multimodal, integrating text, code, images, audio, and structured data, the range of affected tasks and occupations expands further, including creative, scientific, and innovation-oriented work (OpenAI, 2023; Alayrac et al., 2022). This expansion implies that more workers will interact with and be affected by AI, though it does not necessarily imply displacement. Some workers and organizations may use AI to become more productive, create new products and services, and generate new forms of growth. This could lead to new firms, new industries, new jobs, and higher pay (Acemoglu & Restrepo, 2019; Agrawal et al., 2019). The extent, direction, and pace of change all matter: how widespread are AI-driven changes across occupations, how intensive are they within jobs, and how quickly do organizations reorganize as AI systems improve?

These changes can generate aggregate consequences (Acemoglu, 2024b; Trammell & Korinek, 2023). Shifts in employment and wages shape household income, consumption, and demand, with heterogeneous effects across regions, sectors, and demographic groups. In turn, macroeconomic conditions influence firm behavior, investment decisions, and the direction of further technological innovation. Economic and technological change co-evolve. Studies that examine AI's labor impact often have distinct objectives, including measuring AI capability and exposure, estimating causal effects, explaining past outcomes, and forecasting future change. A predictive framing must incorporate these perspectives, since each informs forward-looking assessment of how capability, adoption, and use translate into

changes in tasks, workflows, and labor market outcomes over time. Predicting AI's impact on labor therefore requires specifying what outcomes are being forecast, at what level of aggregation, and over which time horizons, since these choices shape model design, data requirements, and evaluation standards.

## 2.1. Outcomes, Levels of Aggregation, and Time Horizons

To make prediction targets concrete, it is useful to specify the unit of analysis and the horizon over which AI-driven labor changes unfold. At the most granular level, work consists of tasks—discrete activities performed by individual workers. A first set of predictive questions concerns (i) capability: which tasks AI systems can perform at a given level of quality and cost, and (ii) realized use: which tasks are performed by AI in practice. These need not coincide, since realized use depends on adoption frictions, organizational constraints, worker preferences, and complementary inputs. Many tasks are connected into workflows (task sequences), which increasingly map onto agentic systems designed to execute multi-step objectives. This motivates workflow-level prediction targets: which workflows become partially automated, how task composition changes within roles, and how human effort is reallocated toward oversight, verification, or higher-level decision-making.

Above the individual worker, AI reshapes labor through teams and organizations as firms redesign roles, adjust headcount, and modify managerial structures around human–AI collaboration. These micro-level changes aggregate into occupation-, firm-, industry-, and region-level outcomes, including diffusion of adoption, productivity growth, labor demand reallocation, and wage dispersion. Importantly, these outcomes unfold across different horizons: some effects emerge quickly (e.g., task time savings), while others depend on slower-moving adjustments such as organizational redesign, worker reskilling, and general equilibrium (industry, regional, and national level) responses.

**Examples of prediction targets:** Predicting AI's labor impacts involves predicting: (i) task substitution/augmentation, (ii) workflow and role redesign within jobs and occupations, (iii) productivity and quality effects at the worker, team, and firm level, (iv) compensation, hours, and job stability, (v) organizational restructuring and diffusion of adoption across firms and industries, and (vi) aggregate labor market outcomes such as employment, wage growth, and inequality. Across these targets, the core complications are that capability does not imply adoption, adoption is endogenous, effects are heterogeneous, and evaluation must occur under non-stationarity and distribution shift.

# 3. Current Approaches and Obstacles to Predicting AI's Impact on Labor

## 3.1. Current Approaches to Predicting AI's Impact on Labor

A large number of studies examine how automation and AI reshape tasks, occupations, and labor market outcomes. In economics, the task-based model of technological change formalizes production as a bundle of tasks allocated between labor and capital (robots, algorithms) and emphasize displacement (automation), productivity, and reinstatement (new tasks) effects (Autor et al., 2003; Acemoglu & Restrepo, 2019). This framework motivates prediction targets that depend on how AI changes share of tasks performed by labor versus capital, and how task bundles within occupations reorganize over time. The task-based framework is widely used in the current AI-and-labor literature, though the broader tradition also includes structural labor-market models and agent-based computational economics.

Empirically, scholars have operationalized these ideas by constructing task and occupation level AI exposure scores (Frey & Osborne, 2017; Arntz et al., 2016). Recent ML- and AI-specific measures include the crowd-sourced task-based Suitability for Machine Learning index (Brynjolfsson et al., 2018), patent-based measures linking task exposure to labor market outcomes (Webb, 2020), and expert- and model-based mappings of LLM capabilities to occupations that capture potential rather than realized exposure (Eloundou et al., 2023).

Experimental studies of generative AI in workplace settings produce causal evidence on near-term productivity effects and heterogeneity. Field evidence in customer support finds productivity gains concentrated among less-experienced workers, alongside shifts in task composition and potential effects on job quality and retention (Brynjolfsson et al., 2025b). Related experimental studies evaluate impacts in consulting (Dell'Acqua et al., 2026), software development (Peng et al., 2023) and other knowledge work contexts (Noy & Zhang, 2023). These provide credible local estimates but limited guidance about diffusion and general equilibrium labor market outcomes.

Management and organizational research emphasizes that adoption frictions and organizational complements, and emphasizes modeling workflow redesign. The "automation–augmentation paradox," describes automation and augmentation to be interdependent over time and that outcomes depend on organizational choices (Raisch & Krakowski, 2021). The literature also emphasizes human–AI complementarity and the role of uncertainty, tacit knowledge, and context in organizational decision-making (Jarrahi, 2018). A related literature on algorithmic management studies how algorithmic control and monitoring reshape work, expand-

ing "labor impact" to include job quality, autonomy, agency, and governance (Kellogg et al., 2020).

In computer science and machine learning, the most relevant contributions concern measuring and predicting capability and its generalization to real-world tasks. Scaling law work documents predictable relationships between compute/data/model size and loss, motivating partial forecastability of frontier capability trends (Kaplan et al., 2020). Benchmarking frameworks such as HELM stress that model performance depends on scenario and metric choices and that evaluation requires multidimensional measurement (Liang et al., 2022). Recent benchmarks such as SWE-bench target economically relevant software engineering tasks, narrowing the gap between ML evaluation and work-like performance measurement (Jimenez et al., 2024). This connects naturally to ML work on domain adaptation and distribution shift (Wang & Deng, 2018; Quinonero-Candela et al., 2009), and to uncertainty quantification methods such as conformal prediction for coverage guarantees under drift (Angelopoulos & Bates, 2021).

Overall, the literature provides key building blocks: task frameworks, exposure indices, micro-level causal evidence, organizational adoption and change theory, and ML capability evaluation. Integrating them into prediction systems that jointly (i) predict evolving capabilities, (ii) model diffusion and organizational redesign, and (iii) predict labor market outcomes could substantially improve our predictions of AI's labor impact.

## 3.2. Core Technical Obstacles to Predicting AI's Impact on Labor

Despite growing interest and an expanding empirical literature, producing reliable predictions of AI's impact on labor remains challenging due to structural constraints that arise from (i) rapidly evolving AI systems, (ii) complex organizational adoption and redesign, and (iii) endogenous labor market adjustment. This section summarizes the key obstacles and explains why overcoming them requires new approaches, better measurement, and evaluation frameworks.

### 3.2.1. DATA AND MEASUREMENT GAPS

Prediction requires measurable targets, yet many of the most important variables in the AI–labor agenda are difficult to observe. First, the "treatment", i.e., AI capability, is not straightforward to measure (Eloundou et al., 2023; Liang et al., 2022). Unlike classical automation technologies such as industrial robots, which are designed for relatively specific tasks and can be counted and categorized by function (Acemoglu & Restrepo, 2020; Graetz & Michaels, 2018; Chung & Lee, 2023), modern AI systems are general-purpose and software-based, with usage that is fluid across tasks, contexts, and modalities. The relevant unit is often

not "how many AI systems exist," but how they are used across workflows, how reliable they are in a given setting, and how they change human task allocation. Moreover, AI capabilities are evolving rapidly.

Second, adoption, effective use, and usage intensity are difficult to measure. AI can be used for coding, writing, analysis, design, customer service, or coordination, and the same model may be deployed differently across firms or even across workers in the same role. Many usage data are proprietary, inconsistently logged, or embedded in broader IT systems, limiting the feasibility of robust population-level inference.

Third, even labor market outcomes, such as employment, hours, earnings, turnover, job transitions, are measured imperfectly. Government statistics provide helpful information but are often based on surveys with declining response rates, limited detail on tasks and workflows, and substantial publication lags (Meyer et al., 2015; Dominski & Lee, 2025; Lee et al., 2025). Proprietary datasets can offer higher frequency and granular information, but they are typically collected for business purposes and may be selective, difficult to validate, poorly aligned with the outcomes of interest, and expensive. Together, these gaps imply that prediction efforts are frequently forced to rely on proxies (e.g., occupation-level exposure measures, job postings, or surveys of self-reported usage) that introduce substantial noise and weaken predictive validity (Raj & Seamans, 2018; National Academies of Sciences, Engineering, and Medicine, 2025).

### 3.2.2. RAPIDLY EVOLVING AI CAPABILITY

Predicting AI's labor impact involves severe non-stationarity because AI capabilities change rapidly, further complicating measurement (Kwa et al., 2026; Dominski & Lee, 2025). Model performance can improve not only through scale, but through changes in training data, tool use, interface design, alignment methods, retrieval, and the emergence of agentic scaffolding. The set of economically relevant tasks affected by AI therefore expands over time, including shifts from text-centric work toward multimodal and more interactive workflows.

This non-stationarity challenges empirical analysis, since the relationship between automation exposure and labor outcomes will likely evolve. In practice, the mapping from model benchmarks or capabilities to real-world work performance can shift quickly, and new deployment paradigms can create discontinuities in the speed and breadth of impact (Liang et al., 2022; Recht et al., 2019). Prediction in this setting is therefore not simply extrapolating trends, but predicting under changing regimes.

### 3.2.3. FROM CAPABILITY TO REALIZED USE

A third obstacle is that capability does not translate directly into labor substitution or augmentation. Even when AI systems can perform tasks at high quality, their labor market effects depend on adoption decisions and the surrounding organizational context (Bresnahan et al., 2002; Raisch & Krakowski, 2021; Jarrahi, 2018). Realized use is shaped by costs, complementary investments, data availability, governance constraints, managerial beliefs, worker acceptance, and the fit between AI outputs and downstream quality requirements.

Moreover, AI's most important impacts may not come from automating isolated tasks but from reorganizing workflows and reallocating responsibilities across humans and machines. For example, AI tools may reduce the need for certain support functions while increasing demand for oversight, integration, validation, or domain-specific decision-making. Firms may redesign roles and teams around new bottlenecks, changing span of control, headcount composition, team structure, and internal career ladders (Raisch & Krakowski, 2021; Valentine et al., 2024). These workflow-level shifts are difficult to infer from coarse "exposure" measures, yet they can drive labor demand changes even when headline capabilities appear stable (Brynjolfsson et al., 2025a; Valentine et al., 2014).

### 3.2.4. ENDOGENEITY AND REVERSE CAUSALITY

A central technical challenge is that AI adoption and labor outcomes are jointly determined. Adoption is endogenous: firms deploy AI precisely where they expect returns to be highest and where constraints are lowest. Workers may also adopt AI tools when they anticipate benefits to productivity, evaluation metrics, or job security. As a result, observed correlations between AI use and productivity or employment outcomes are not easily interpreted as causal, and models trained on observational data may learn patterns that do not hold under counterfactual conditions.

Reverse causality further complicates prediction. Labor shortages, wage growth, or shifting demand conditions can themselves drive AI investment, meaning that labor market conditions shape AI adoption as much as AI adoption shapes labor market conditions. These bidirectional dynamics weaken the reliability of forecasts that treat AI adoption as an exogenous input, and motivate predictive approaches that explicitly model selection and feedback (Athey & Imbens, 2017).

### 3.2.5. HETEROGENEITY AND AGGREGATION ERRORS

AI's labor impacts are heterogeneous across tasks, occupations, firms, industries, and regions, and this heterogeneity creates both scientific and practical obstacles for prediction (Autor & Thompson, 2025; Dell'Acqua et al., 2023). Within the same occupation, workers differ in task mix, experience, complementary skills, and access to tools. Across firms, differences in managerial practices, IT infrastructure, data availability, and organizational culture can produce widely varying productivity gains and substitution patterns from the same underlying AI capability.

These differences interact with a persistent unit-of-analysis mismatch. Many widely used measures operate at the occupation level, while AI systems act at the task and workflow level, and organizational redesign occurs at team and firm levels. Aggregating task-level exposure into occupation-level risk can therefore produce systematic errors, over-predicting displacement in some jobs and underpredicting reallocation and redesign in others. More broadly, averaging heterogeneous effects can obscure distributional consequences, including changes in wage dispersion, promotion paths, and employment risk for specific subgroups.

### 3.2.6. GENERALIZATION AND EVALUATION UNDER DISTRIBUTION SHIFT

Evaluation is also difficult. Many economically meaningful outcomes only unfold over years, and the statistics that measure them, such as employment, wages, or job transitions, are revised repeatedly as more complete data arrives. A forecast made today about how AI will reshape clerical work, for instance, may not be properly evaluated for several years, and even then the relevant numbers may still shift. Targets are also multidimensional. A model might predict aggregate employment accurately while missing what happens to wages, hours, or job quality, or it may track average trends while missing that gains and losses concentrate among specific workers or regions. Distributional and tail outcomes can matter more than averages, since a 10% chance of an occupation contracting for college graduates carries very different policy implications from a small average change. Prediction frameworks must therefore be judged not only on point accuracy but also on robustness across settings and calibrated uncertainty, especially in high-stakes contexts where overconfident forecasts can misguide policy or drive sharp market reactions.

## 4. Research Directions for Predicting AI's Impact on Labor

This section outlines a research agenda for overcoming the obstacles identified in Section 3. The suggestions are illustrative rather than exhaustive. The broader takeaway is that the AI and ML community can play an active role in shaping this interdisciplinary research agenda.

## 4.1. Data and Measurement Infrastructure

Prediction is constrained by the fact that standard labor market datasets were not designed to measure AI-mediated work. Official statistics are often high-quality but lagged and limited in task detail, while proprietary datasets can be timely but selective and optimized for business purposes rather than labor measurement (Lee et al., 2025; Raj & Seamans, 2018; Meyer et al., 2015).

A central direction is therefore to build measurement pipelines that track task and workflow change and link them to AI use. In an ideal setting, prediction would rely on longitudinal measurement of task allocation and workflow composition, AI tool usage and intensity, worker skills and training, productivity and quality outcomes, and firm-level deployment and redesign decisions. While such data rarely exists in full, progress can come from two complementary strategies: (i) creating new measures of AI exposure, capability, and usage that reflect workflows rather than static occupational categories, and (ii) extracting structured signals from existing unstructured data sources (e.g., job postings, company reports, work artifacts, logs, and text descriptions of tasks) (Hershbein & Kahn, 2018; Hassan et al., 2019; Li, 2010).

A particularly promising approach is ensemble measurement, where multiple imperfect indicators (surveys, postings, usage statistics, administrative records, and text-based proxies) are combined and cross-validated. This reframes measurement as a prediction problem in itself, with the goal of constructing a robust ensemble measure of "AI use."

## 4.2. Benchmarking Tasks and Workflows

Because AI capabilities evolve rapidly, predicting labor impacts requires evaluation frameworks that measure what systems can do in work-like settings (Liang et al., 2022; Jimenez et al., 2024). A distinctive contribution ML can make is to help build economically grounded benchmarks aligned with occupational tasks and workflows, including multi-step objectives, tool use, interaction, and performance evaluation. The software engineering case is the clearest existing example (Jimenez et al., 2024), and recent work extends it from isolated tasks toward realistic multi-step workflows and tasks with measurable economic value (Xu et al., 2025; Miserendino et al., 2025).

Domain experts, including economists, organizational researchers, and industry practitioners, should lead benchmark construction and specify what constitutes realistic task completion, acceptable quality, and meaningful failure. The contribution of ML researchers would be benchmarking infrastructure and practice such as versioned evaluation sets and reproducible scoring protocols. SWE-bench works because practicing engineers recognize the tasks as genuine

(Jimenez et al., 2024). The same criterion must govern other occupational benchmarks. Benchmark design requires collaboration in which domain experts and ML evaluation engineers each inform the other.

One direction is occupation-level workflow benchmarks, where a benchmark suite represents realistic work sequences (e.g., producing reports, coordinating stakeholders, validating outputs) rather than isolated tasks. A second direction is team- and organization-level benchmarks, where multiple agents interact in roles resembling organizational structures, for example independent contributor agents operating under managerial oversight and verification (Xu et al., 2025). In both cases, evaluation should go beyond completion rates to include quality, error modes, oversight requirements, and cost.

## 4.3. Prediction Methods for Adoption, Drift, and Feedback

Even with improved measurement and benchmarking, prediction requires methods that explicitly address non-stationarity, endogenous adoption, and feedback. A central objective is to develop prediction frameworks that jointly model AI capability growth, adoption and effective use, organizational and workflow adjustment, and downstream labor market outcomes. Integrating these elements within a single empirical framework helps clarify where uncertainty enters and which intermediate outcomes, such as adoption or task reallocation, can be evaluated and updated, even when long-horizon aggregate effects remain uncertain.

This framing motivates prediction approaches that go beyond static extrapolation of past trends, and several families of ML methods are designed for precisely the regime-change setting that AI-driven labor disruption creates. Continual and online learning frameworks update models as new data arrive rather than assuming a fixed training distribution (Parisi et al., 2019), which matters when the set of AI-affected tasks expands with each model generation. Distributionally robust optimization stress-tests predictions against worst-case shifts across subgroups, such as the firms, sectors, or regions where adoption diffuses unevenly (Sagawa et al., 2020), guarding against forecasts calibrated only to early adopters. Conformal prediction supplies calibrated predictive intervals with finite-sample coverage guarantees, including recent variants that retain validity under distribution shift (Angelopoulos & Bates, 2021; Barber et al., 2023), so that forecasts carry honest uncertainty when the regime is changing rather than overconfident point estimates. We highlight these methods because they address a limitation that standard econometric tools were not designed for: prediction when the data-generating process itself is moving.

For policy-relevant questions, calibrated intervals and sensitivity to tail risks are often more informative than point

forecasts (Ovadia et al., 2019). Finally, hybrid approaches that combine structural modeling, causal inference, and machine-learning prediction are particularly well suited to this domain, where adoption and labor outcomes are jointly determined and generalization beyond historical experience is the primary concern (Athey & Imbens, 2017).

### 4.4. Simulating Labor Market Responses with LLM-Based Agents

An emerging direction uses large language models themselves as simulated economic agents. Because LLMs are trained on vast records of human reasoning and behavior, they can be conditioned on personas, endowments, and constraints and then queried as stand-ins for economic agents (Park et al., 2023; Horton et al., 2026). LLM agents endowed with preferences and information reproduce qualitative findings from classic behavioral and economic experiments (Horton et al., 2026). Chopra et al. (2025) integrate LLM-based agents into agent-based models at the scale of millions, simulating the labor-force participation of 8.4 million agents in a digital twin of New York City and validating aggregate behavior against census and labor statistics. Karten et al. (2025) embed census-calibrated worker agents in a two-level reinforcement learning game with a planner agent, using in-context learning to design tax policy and showing that population-scale simulation can be coupled directly to mechanism design.

For predicting AI's impact on labor, this paradigm can be attractive. It can represent heterogeneous behavior at population scale, capturing how workers and firms with different skills, constraints, and beliefs might respond to AI-driven changes, which representative-agent models cannot easily do. Also, simulation can explore counterfactual regimes that have not yet been observed. Where empirical estimates are bound to the institutional and technological environments present in historical data, simulated agents can be placed in hypothetical futures and their collective behavior examined before such a regime arrives. However, this method comes with serious caveats. Simulated behavior can reflect the idiosyncrasies and biases of the underlying model rather than the population of interest. Results will likely be sensitive to prompt design and persona construction.

At this stage, LLM-based simulation is most likely a complement to existing methods rather than a replacement and could be helpful for generating and stress-testing hypotheses about labor market responses. The field is relatively new and greater involvement from social scientists and from the ML community could sharpen the methods and their validation considerably.

### 4.5. Evaluation Design

Predicting AI's labor impact requires shared standards for evaluating predictions, not only for measuring capabilities. Individual studies provide valuable local evidence but cannot easily accumulate when they use different targets, horizons, and datasets. Like randomized controlled trials moved from isolated experiments toward standardized, scaled-up designs, AI–labor prediction needs shared evaluation standards that facilitate comparability, robustness, and calibration under change.

Useful practices include scoring forecasts repeatedly over time and at different horizons, breaking results down by occupation, region, and demographic subgroup, and checking how models perform when conditions change sharply, such as after major shocks or as AI capabilities advance. Scoring should explicitly account for uncertainty using proper scoring rules such as the Brier score, which penalize models that are confidently wrong rather than only those that are inaccurate on average.

A practical step is to create shared prediction tasks and competitions in which teams use common data and rules to forecast outcomes such as occupation-level employment changes, wage trajectories, task reallocation, or adoption rates, scored as the outcomes are realized. The value of such competitions depends on who designs them. The most informative challenges, in the spirit of Kaggle, would be those where labor-market specialists define the prediction targets and success criteria and methods compete on that common ground. Such competitions can convince experts of a modeling approach's value, and over time they would accumulate into shared prediction tasks, like ImageNet did for computer vision.

### 4.6. Institutional Design and Incentives for an Interdisciplinary Labor Prediction Ecosystem

Whether this agenda becomes a sustained research program depends on institutional incentives. Several steps can help. The most direct is to legitimize AI–labor prediction as a research direction at leading venues in each discipline. ML venues such as ICML could support dedicated tracks, workshops, and shared tasks. Economics and management venues should encourage work that brings ML methods and evaluation standards on labor questions. Papers or sessions proposing improved labor market prediction must ultimately convince the disciplines that study the topic. Furthermore, a signal from a venue like ICML that labor prediction is a legitimate ML problem carries weight across fields and helps shape what funding agencies prioritize. Grants for studying AI's labor impact are largely directed at economists using established designs such as randomized controlled trials. Demonstrating that ML–economics collaboration produces results can incentivize foundation and industry to prioritize

and fund research on the topic.

Sustaining scientific rigor across this collaboration will be important. Interdisciplinary research risks impressing one field while resting on naive assumptions from another, for instance standard economics dressed in ML vocabulary, or the reverse. Co-authorship across disciplines and review by both communities would safeguard against so-so interdisciplinary work and help develop novel research. Collaboration with industry will also be valuable, since real-time information about AI use, workflow redesign, and productivity exist inside firms. Voluntary disclosure of proprietary data is unlikely, and hence private-public-academia partnership that recognizes the importance of examining AI's labor market impact and methods to ensures data privacy will be critical

Finally, prediction must be global in scope. AI development is concentrated in a handful of firms in a handful of countries, but labor impacts will spread globally and vary across regions and countries. As such, evaluation settings should include diverse contexts, including low- and middle-income countries, and avoid generalization from early-adopting, high-income settings (OECD, 2023; Melina et al., 2024).

## 5. Alternative Views

This section situates the paper's position against existing prediction paradigms in economics, then addresses two alternative views: that prediction is unnecessary because markets will adjust, and that it belongs within economics rather than machine learning.

### 5.1. Prediction paradigms in economics

Beyond exposure indices and reduced-form estimates, a long tradition in macro and labor economics uses structural models to study technology shocks, diffusion, and equilibrium adjustment. These include search-and-matching frameworks (Mortensen & Pissarides, 1994), DSGE-style models with labor market frictions, and models of firm heterogeneity (Hopenhayn, 1992). Related approaches include agent-based models of technology diffusion and organizational adaptation (Farmer & Foley, 2009). Structural labor-market models provide equilibrium interpretations of technology shocks, yet their quantitative predictions can be fragile and highly sensitive to modeling assumptions (Shimer, 2005). To date, the application of these structural approaches to AI remains relatively scarce (Wang & Wong, 2025). In contrast, much of the empirical literature relies on reduced-form panel and time-series methods designed for staggered adoption and dynamic and heterogeneous treatment effects (Sun & Abraham, 2021; Callaway & Sant'Anna, 2021). These approaches are well suited to counterfactual policy analysis, but are challenging to extend to long-horizon equilibrium predictions. At the macroeconomic level, forecasting has

traditionally relied on linear time-series models, including structural VARs used to identify technology shocks (Blanchard & Quah, 1989). While these models perform well under stable dynamics, their performance deteriorates under structural change and evolving technologies (Stock & Watson, 2012). More recently, machine-learning methods have been incorporated into economic forecasting to handle high-dimensional predictors and nonlinearities (Varian, 2014; Athey & Imbens, 2019). These approaches can improve short-horizon accuracy, but they remain fragile under non-stationarity and are limited in their ability to support policy counterfactuals. Across these methods, i.e., structural models, reduced-form panel methods, and macro time-series models, applications to predicting AI's labor impacts are often constrained by measurement of AI capability and use, strong functional-form assumptions, and limited evaluation under non-stationarity and dynamic change. There are approaches that can help mitigate these challenges. Quasi-experimental methods identify heterogeneous and dynamic treatment effects, and structural models represent regime transitions explicitly. The more precise limitation is that their estimates are grounded in observed regimes, so extrapolation becomes difficult when the set of tasks and occupations themselves are shifting rather than the parameters governing existing ones. ML methods also rely on observed data and face their own extrapolation limits. The point of this paper is to promote complementarity. Economics is optimized for variation within an observed institutional and labor market structure and identifying causal effects, ML is optimized for flexible high-dimensional modeling and detection of distribution shift when that structure may be changing, and LLM-based simulation for exploring regimes not yet observed (Horton et al., 2026; Chopra et al., 2025; Karten et al., 2025). Integrating both approaches most likely would enable a richer and more reliable labor-impact prediction.

### 5.2. Prediction is unnecessary because markets will determine outcomes

One alternative view holds that predicting AI's labor impacts is unnecessary or misguided, because competitive markets will determine efficient technology adoption, task allocation, and occupational adjustment without the need for ex ante prediction or intervention. From this perspective, long-horizon forecasts are either infeasible under rapid technological change or irrelevant, as prices, wages, and organizational forms will adjust endogenously. This view echoes the laissez-faire tradition in economics that is skeptical of planning and expert forecasting. It warns against forward-looking analysis followed by heavy-handed policy intervention.

This view is not unfounded, but the argument here is not for heavy-handed intervention. Prediction does not necessarily

go against markets, but in fact prediction can facilitate better information. Market participants themselves require forecasts to act: firms decide whether to invest in AI, workers whether to reskill, and investors price firms on expected disruption, all on the basis of expectations about future impacts. Accurate prediction improves these decentralized decisions rather than displacing them. Even efficient long-run adjustment can involve costly transitions, with concentrated displacement, skill mismatch, and distributional consequences that forward-looking analysis can help anticipate. The alternative to rigorous prediction is not the absence of forecasts but their sensational, and often inferior, non-peer-reviewed substitutes, which can distort markets (The Wall Street Journal Staff, 2026).

### 5.3. Labor market prediction belongs in economics, not ML

A second perspective accepts that predicting AI's labor impacts is valuable, but argues that this task should remain primarily within economics. From this view, existing structural and reduced-form methods are seen as sufficient for examining causal impacts and general equilibrium effects of AI on then economy. Machine learning is best considered as techniques that excel in data construction and measurement, and one that solves a distinct problem often labeled as the $\hat{y}$ problem rather than the $\hat{\beta}$ problem.

However, forward-looking scientific prediction of labor markets can better prepare societies to anticipate and adapt to technological change. Modern ML and LLM-based frameworks can complement economics-based causal and structural estimation by requiring explicit benchmarks, robust evaluation under distribution shift, calibration under change, and conducting LLM-based simulations.

## 6. Conclusion

AI is rapidly reshaping how work is performed. A key challenge is predicting how evolving AI capabilities interact with adoption, organizational change, and political economic adjustment to produce shifts in the labor market, including tasks, employment, wages, and inequality. This paper has argued that this challenge should be treated as a core machine learning problem, beyond an AI policy or ethics question. Predicting how evolving AI systems reshape work requires prediction under non-stationarity, distribution shift, endogenous feedback, and high-stakes uncertainty, challenges that lie at the center of modern machine learning, but also problems that economics have long tackled.

Addressing these challenges will require moving beyond isolated disciplinary studies toward a cumulative prediction science: clearly defined prediction targets, scalable measurement of AI use and task change, evaluation under

regime change, and calibration under change. The ML community can contribute in distinctive ways, particularly in benchmarking, robust prediction, LLM-based simulation of economic behavior, and evaluation design. Innovation can happen through sustained collaboration with economists and other social scientists, pursued in machine learning venues as well as economics and management venues, alongside shared datasets and recurring evaluation tasks. As AI systems increasingly perform economically valuable work, predicting their labor impacts will remain central to machine learning's trajectory, since each advance in the technical frontier reshapes the economic work it performs.

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
