# OpenReview forum: "Position: Predicting AI’s Impact on Labor Is a Core Machine Learning Problem"
_ICML.cc/2026/Position_Paper_Track — ICML 2026 Position Paper Track regular_

### Official Review · Reviewer_PPhw · 2026-03-12

**Significance:** 2
**Argument Clarity:** 3
**Rating:** 5
**Confidence:** 2

**Questions:**

none

**Alternative Views Section:**

Yes

**Compliance With Llm Reviewing Policy A Conservative:**

Affirmed.

**Discussion Potential:**

2

**Final Justification:**

Improving my score from 3 to 5; the authors have made some points much more clearer and promised to weaken some statements that I found too strong initially. I'm also now of the opinion that this particular sort of work can induce general interest within certain sections of the ICML audience who might be intrigued by the problem type represented by a task like predicting labor market effects.

**Paper Summary:**

The submission calls for greater involvement and research agenda-shaping from the ML research community towards predicting how AI will influence society at large, particularly the economic effects. Aspects associated with the challenge such as non-stationarity, endogenous feedback in systems, distributional shift, and uncertainty modelling are identified as areas that ML research concerns itself with, therefore, ML research likely has things to contribute, such as the culture of benchmarking with evaluation sets and general aspects of prediction science.

**Position:**

Yes

**Position In Title:**

Yes

**Related Work:**

3

**Strengths And Weaknesses:**

Caveat: I’m not familiar at all with the literature in this space, most of which I imagine appears in labour economics and sociology journals, so this review will be mostly naive.

Strengths:

The submission’s overall call to a greater interdisciplinary approach towards assessing the role of evolving AI in the immediate economic and societal future seems worth consideration. The manuscript is well-expressed and it was easy to read.

Weaknesses:

In the sections that cover background and problem-characteristics, it’s unclear to me as a non-expert if  there’s an undue emphasis on a specific viewpoint from economists (e.g. the task-based view of productivity) and if there are other perspectives in the literature that are taken equally seriously by the academic community in economics/social-sciences, but aren’t covered with adequate balance here. It made me wonder if such articles are well-targeted towards the ICML community, where I imagine most readers do not have background to assess the characterizations put forward, and if the real challenge here is convincing the other side that ML research has something meaningful to offer. Of course, both sides need to be encouraged.

Section 4 presents aspects of such research where ML might play a role.
* Here too, e.g. in section 4.3, it’s not clear to me that “A distinctive contribution ML can
make is to develop economically grounded benchmarks aligned with occupational tasks and workflows, including multi-step objectives, tool use, interaction, and performance evaluation.”; this feels very much alien territory for the field, and very likely to lead to over-simplifications.
* Section 4.5 in fact includes a very good call-out in this section to RCTs: RCTs didn’t come from ML folk, they came from the statistical community. It makes me nervous, as someone who reviews ML papers, to incentivize ML researchers to start developing benchmarks aimed at convincing ML-conference reviewers, which bring me to:
* Section 4.6 encourages interdisciplinary work with ML with special tracks at ML venues; in my opinion, this is a little backwards. For one thing, a paper claiming to bring improved modelling through ML in a separate field, such as economics or statistics or medicine, need not convince the ML community but rather the specialized community. For another, it would be very hard to get a reviewing system going, case in point: I’m reviewing a paper that heavily cites work I have zero familiarity with.
* Regarding ML methods of prediction-science, my impression is that while there is literature, ML has not really shown any promise so far in challenging, highly data-starved, even “one-shot” problems such as the ones tackled in some of the economic/scientific fields. Economists and social scientists already use statistical techniques, sometimes more sophisticated ones than the ones ML folk use (for example, our papers rarely report proper measurement of confidence intervals).

**Support:**

1

---

> ### Author Rebuttal · Authors · 2026-03-30
>
> We thank Reviewer PPhw for the frank and detailed comments. These pushed us to think more concretely about our arguments. We resonate with the skepticism about interdisciplinary collaboration advocated without concrete strategies. Calls for interdisciplinarity often go unmatched by the disciplinary incentives. The reviewer's point about the directionality of interdisciplinary diffusion is well-taken. We address the comments below and hope to convey greater optimism for the value of this collaboration.
>
> On disciplinary balance. The task-based framework is dominant in the AI-and-labor literature, and we use it because it maps naturally onto ML concepts of tasks and capabilities. But it is one lens among several, including structural labor market models (search-and-matching, DSGE), agent-based computational economics, and institutional perspectives emphasizing power and governance. We will describe these in the revision. The interdisciplinary collaboration we advocate is aimed at integrating diverse frameworks, not having one dominate.
>
> On Section 4.3 (benchmarking). The reviewer suggests this is "alien territory." We think the coding domain provides a direct counterexample. SWE-bench (Jimenez et al., 2024) measures AI on realistic software engineering workflows, not isolated subtasks. It works because ML researchers understand the full workflow: how code is written, reviewed, tested, and integrated. This workflow-level understanding enables realistic capability assessment with direct labor market implications. TheAgentCompany (NeurIPS 2025) and SWE-Lancer (2025) extend this to broader professional tasks with real economic value.
>
> On benchmark incentives (Section 4.5). The reviewer's concern is well-taken: benchmarks designed to impress ML reviewers risk becoming self-referential rather than scientifically useful. The benchmarks we envision are not internal ML leaderboards but joint constructions where domain experts define what counts as realistic task completion, acceptable quality, and meaningful failure modes. SWE-bench works not because it satisfies ML reviewers but because practicing software engineers recognize the tasks as genuine. The same principle should govern extension to other occupations: if economists and organizational researchers do not find the benchmarks credible, they serve no predictive purpose. We will add this design criterion explicitly in the revision.
>
> On Section 4.6 (why ICML). We agree the argument must be bidirectional. But economics is actively adopting ML methods (Athey & Imbens, 2019; Chernozhukov et al., 2018), and a signal from ICML that labor prediction is a legitimate ML problem carries weight across fields and shapes what funding agencies prioritize.
>
> On what ML specifically adds. Economics excels at causal identification under stable conditions. But AI-driven labor change violates the stationarity assumption. ML offers conformal prediction under distribution shift for calibrated uncertainty when the regime is changing, and distributional robustness methods (Sagawa et al., 2020) for stress-testing against worst-case shifts. These address a structural gap in econometric methods. The reviewer's RCT analogy is instructive: RCTs were developed in statistics, adopted by medicine, and eventually transformed development economics. We envision similar cross-pollination, where ML's evaluation infrastructure is adopted for labor prediction, creating cumulative standards that economics currently lacks.
>
> On the venue question: There are research that already sits at the ML-economics intersection, such as LLM-based simulation of economic agents (Horton et al. 2026). Chopra et al. (AAMAS 2025) scale to millions of agents validated against census data. Karten et al. (2025) integrate RL with economic mechanism design. These use ML methods to address economic questions, and ICML is well-positioned for such work.
>
> On data-scarce settings. This is a fair concern but "data-starved" is partly a measurement problem, not a fixed constraint. Section 4.2 proposes new infrastructure for data construction. Also, some sub-problems are not data-scarce at all: capability benchmarks generate large evaluation volumes, job postings number in millions, usage logs provide high-frequency signals. The scarcity is concentrated in specific components (organizational restructuring, long-run equilibrium), but the binding data constraints are narrower than the reviewer's framing suggests
>
> Our planned revisions are: (1) expand alternative frameworks discussion, (2) add concrete ML examples (agent simulation, prediction under shift, benchmarks), (3) articulate ML's distinctive contribution as addressing a binding constraint rather than claiming disciplinary sufficiency, (4) specify that labor-prediction benchmarks must be co-constructed with domain experts and validated by the target disciplines, not optimized for ML-venue acceptance alone, and (5) strengthen the bidirectional framing of the ICML venue argument.

---

> > ### Author Rebuttal · Reviewer_PPhw · 2026-04-03
> >
> > Thanks to the authors for their rebuttal. I have a few further comments:
> >
> > * SWEBench is indeed a good example precisely for the reason the rebuttal identifies: ML developers work with code, with mostly people who have background in CS and are therefore adequately familiar how code works. My initial point is not refuted by this "counterexample", in fact it's an indirect supporting argument.
> >
> > * I am still unconvinced that the ML community should have much to say about benchmarking; my naive assumption would be that the economics community is much better placed to develop practical benchmarks that simulate real-life conditions, and such benchmarks must be reviewed by the economics community for validation. The ML community can of course then leap into throwing ML methods to climb the leaderboards, and show that the ideas we've been discussing, e.g. objects appearing in different backgrounds and causing OOD issues for vision classifiers, actually do work on problems from other fields. The audience to convince would remain the economics community. However, I do concur with the perspective that there is value to signalling that "labor prediction is a legitimate ML problem carries weight across fields" and that it would "shape what funding agencies prioritize". The only thing to watch out for is incentivizing poor scholarship and lack of rigour.
> >
> > * "Economics excels at causal identification under stable conditions." --> I am once again unable to assess the truth of this statement due to lack of expertise. However, I would naively guess that it is unlikely that an entire academic field is unable to account for unstable conditions; it feels as though it must be the case that real-life instability is in fact very very hard to deal with, maybe even impossible to do so with non-vacuous bounds (e.g. chaotic non-linear dynamical systems), hence the current approaches of regularization through simple models and strong priors on distributions.
> >
> > * Regarding the things that ML offers, while conformal prediction is quite useful for better quantifying uncertainty, the distributional robustness literature almost universally deals with artificial simplistic scenarios that have never been stress-tested in hard real-life scenarios. In my opinion, proving the utility of these methods is best conducted by initially setting out goals by labor economy specialists, and then ML methods should attempt to show that these approaches win. For example, there are numerous examples of Kaggle competitions being set up by specific domain specialists with great care, and then the winning entry from a deep learning method has convinced the community of the value of the modelling approach.
> >
> > Overall, the proposed changes sound good to me, it seems like they would add a more balanced take.

---

### Official Review · Reviewer_G2pV · 2026-03-16

**Significance:** 3
**Argument Clarity:** 3
**Rating:** 5
**Confidence:** 3

**Questions:**

See Weaknesses

**Alternative Views Section:**

Yes

**Compliance With Llm Reviewing Policy A Conservative:**

Affirmed.

**Discussion Potential:**

3

**Final Justification:**

Post rebuttal, I believe the proposed changes will strengthen the manuscript. I increase my score from 4->5 accordingly.

**Paper Summary:**

This paper argues for the position that predicting the impacts of AI on labor is a core machine learning prediction problem. Framing it as an end to end prediction problem, the paper first lists the possible prediction targets of interest in understanding the impact at both micro and macro levels. The paper then highlights the inherent underlying complexity for solving this, followed by research directions that could help alleviate these challenges, and concludes with a discussion on alternative views.

**Position:**

Yes

**Position In Title:**

Yes

**Related Work:**

3

**Strengths And Weaknesses:**

Strengths:

- The paper is well written and easy to follow, and the contribution is timely.
- The position appears novel, is of real significance and is articulated cleanly
- The paper is very comprehensive in its coverage, in terms of identifying the associated challenges as well as how to go about solving them. The paper also engages well with the alternate viewpoints.


Weaknesses:

- My main concern is that the position is too optimistic, in terms of:
   - Feasiblity: The problem of predicting impact on labor is just too hard and that no systematic approach is likely to yield accurate predictions (see very recent relevant work that argues against it in [1]). The paper does not offer concrete technical evidence to counter this skepticism, and better engaging with this view would strengthen its claim.
   - Scope: The paper frames this as a core ML problem but call to action involves dealing with challenges well beyond the ML related predictive aspects, such as understanding adoption dynamics, wage effects, organizational restructuring, and corresponding policy responses. I am also not convinced that just pushing this through ML avenues alone, or relying on good faith industry participation (especially given that there is active tension around the optics of the issue) can yield the requisite interdisciplinary outcomes.

- Typo: Missing Reference on Line 326

Nevertheless, I feel positive about the paper’s timely contributions and I am willing to improve my opinion if my concerns are addressed.

References:\
[1] Can We Predict What Jobs AI Will Take? Davenport and Paredes, 2025

**Support:**

3

---

> ### Author Rebuttal · Authors · 2026-03-30
>
> We thank Reviewer G2pV for the thoughtful engagement and for recognizing this piece as a potentially novel and timely contribution.
>
> 1. On Feasibility. First of all, thank you for pointing us to Davenport & Paredes (2025). We found the piece valuable. Their diagnosis of the literature is in line with our assessment: existing predictions failed because they relied on coarse occupation-level scores, lacked ground-truth data, and produced sensational point estimates. Our Section 3 identifies similar problems: measurement gaps (3.2.1), non-stationarity (3.2.2), the capability-adoption gap (3.2.3), and evaluation failures (3.2.6). We also recommend improving measurement infrastructure (4.2).
>
> However, the difficulty of predicting AI's labor market impacts does not imply that prediction is speculative. The critique that forecasting monolithic (and often sensational) outcomes like the number of jobs destroyed or created is futile is precisely why a decomposed, methodologically rigorous approach is needed. We will make this point clearer in Section 1. The question is whether individual sub-problems are tractable, and there is growing evidence that they are.
>
> For example, scaling laws (Kaplan et al., 2020) demonstrate that capability forecasting is partially tractable, and benchmarks like SWE-bench, TheAgentCompany (NeurIPS 2025), and SWE-Lancer (2025) now track performance on economically relevant tasks across model generations. On adoption and usage, the Anthropic Economic Index provides revealed-usage data that the Yale Budget Lab has already used to assess early employment patterns. On productivity effects, experimental evidence (Brynjolfsson et al., 2023; Dell'Acqua et al., 2023) provides calibration targets.
>
> Also, LLM-based simulation of economic behavior is emerging as a powerful tool for modeling how workers and firms might respond to AI-driven changes. Horton et al. (NBER, revised 2026) demonstrate that LLM agents reproduce classic experimental results. Chopra et al. (AAMAS 2025) validate population-scale simulations against census data. Karten et al. (2025) use reinforcement learning for economic mechanism design with heterogeneous agents. These are novel ML-economics collaborations producing evaluable results.
>
> The stakes of getting this right are concrete: recently, a non-peer-reviewed scenario analysis (Citrini Research) moved stock markets. This illustrates precisely why transparent, methodologically rigorous prediction infrastructure is needed to counter sensational reports and ground the study of AI's labor market impact in scientific standards.
>
> 2. On Scope. We agree that predicting AI's labor market impact requires perspectives from economics, management, and the social sciences. But we believe the research agenda is fundamentally interdisciplinary, both methodologically and because the technology itself is being developed in real time, explicitly aimed at performing human economic work. Prediction will span capability, adoption, organizational redesign, wages, and governance, and progress requires not only methodological collaboration but intentional co-design across ML, economics, management, and adjacent social sciences.
>
> Our key claim is that ML contributions resolve key constraints. Without credible capability forecasting and evaluation infrastructure, adoption models may lack realistic inputs and policy analysis will be speculative. We target an ML audience because that is where the constraint sits and because this community is building the technology whose impacts are in question. We will make this distinction crisper in Section 1, and we will foreground the interdisciplinary scope earlier in the paper.
>
> The concern about industry incentives is quite valid. We do not assume good-faith participation. Some useful data is already public (e.g., the Anthropic Economic Index), and privacy-preserving methods can reduce reliance on voluntary disclosure. But we also see value in making the ask explicit at an ML venue: this community has established data-sharing and evaluation norms before (ImageNet, SWE-bench), and it can do so here. The alternative, ceding the forecasting space to non-peer-reviewed industry reports serves no one well, not workers, firms, or policymakers.
>
> Thank you for pointing out the missing reference (Line 326): We have fixed this now.
>
> In sum, our planned revisions are: (1) engage Davenport & Paredes in Alternative Views, (2) add concrete ML examples (agent simulation, prediction under non-stationarity, benchmarks), (3) clarify that ML contributions are a binding constraint on the broader interdisciplinary agenda rather than a claim of disciplinary sufficiency, (4) address data access constraints and institutional incentives for disclosure, and (5) propose concrete cross-disciplinary mechanisms, including shared prediction tasks and interdisciplinary workshops designed to bring economists and social scientists into the evaluation infrastructure from the outset.

---

> > ### Author Rebuttal · Reviewer_G2pV · 2026-04-03
> >
> > I am happy with the proposed changes; I increase my score accordingly. I also urge the authors to give serious consideration to Reviewer PPhw's suggestions.

---

### Official Review · Reviewer_942X · 2026-03-22

**Significance:** 2
**Argument Clarity:** 3
**Rating:** 5
**Confidence:** 3

**Questions:**

1. Can you provide more concrete examples of how machine learning techniques can be applied to predict AI's impact on labor? Are there specific models, datasets, or case studies that you can discuss to illustrate the potential of ML in this area? Any simple experiments or preliminary results would be helpful to support your position.
2. What are some potential limitations or ethical considerations of using machine learning to predict AI's impact on labor? How do you propose or call for action to address these issues in the research?
3. How do you see the role of interdisciplinary collaboration in this area? For instance, how can machine learning researchers work with economists, sociologists, and policymakers to better understand and predict the impact of AI on labor?

**Alternative Views Section:**

Yes

**Compliance With Llm Reviewing Policy A Conservative:**

Affirmed.

**Discussion Potential:**

3

**Final Justification:**

The authors answered my questions well and addressed my main concerns regarding concrete examples of machine learning models used for the topic and the call for action for collaboration between machine learning experts and economists. I’m happy to adjust my score.

**Paper Summary:**

This paper argues that predicting the impact of AI on labor is a core machine learning problem rather than merely a social or ethical issue. The authors claim that understanding and forecasting the effects of AI on jobs, wages, and economic structures requires sophisticated machine learning techniques to analyze complex data and model the interactions between AI technologies and labor markets. They also discuss key challenges and prediction tasks in this area.

**Position:**

Yes

**Position In Title:**

Yes

**Related Work:**

3

**Strengths And Weaknesses:**

The paper has a clear position and is supported with some reasoning and evidence. The authors provide a comprehensive analysis of what needs to be done to predict AI's impact on labor. From there, the importance of machine learning in addressing this challenge is naturally revealed. The topic of AI's impact on labor is highly relevant and important to the ICML community, as it has significant implications for the future of work and society. The paper is likely to inspire discussion, as it challenges the common perception that this issue is more a social or economic one with ML merely being a tool. The paper is clearly argued and cites related work appropriately.

One suggestion for improvement is to provide more concrete examples of how machine learning techniques can be applied to predict AI's impact on labor. For instance, the paper could discuss specific models, datasets, or case studies that illustrate the potential of ML in this area. Additionally, the paper could also address potential limitations or ethical considerations of using ML for this purpose.

**Support:**

3

---

> ### Author Rebuttal · Authors · 2026-03-30
>
> We thank Reviewer 942X for the helpful comments and supporting the idea that AI's impact on labor is important and relevant to the ICML community.
> 1. Concrete Examples: We agree and will add a dedicated discussion of the below examples in Section 4.
>
> (a) LLM-based simulation of economic behavior: Recently there has been several papers using LLMs as simulated economic agents. Horton, Filippas & Manning (NBER, revised 2026) show that LLM agents endowed with preferences and constraints reproduce classic economics experimental results, with treatment effects correlating at r=0.85 with human subjects. Chopra et al. (AAMAS 2025) scale a similar exercise to 8.4 million agents in a digital twin of New York City, validated against census data on labor force participation and mobility. Karten et al. (2025) combine census-calibrated worker agents with reinforcement learning to optimize tax policy, framing the planner-worker interaction as a two-level RL game. These approaches represent a new way to simulate heterogeneous economic behavior at population scale, with ML and AI concepts and methods (persona-conditioning, in-context learning, RL for mechanism design, scalable architectures) that are essential to making the approach work. For labor prediction, this opens the possibility of simulating how workers and firms respond to AI-driven changes under different scenarios, something that traditional economic models can only do under highly stylized assumptions. These approaches are still in early stages with a relatively small community of ML scholars involved. More interest and involvement from the ML community could spur more diverse and rigorous approaches to AI-based simulation methods.
>
> (b) Prediction under non-stationarity: A key feature of the AI-labor prediction problem is that the data-generating process itself changes as AI capabilities evolve. Standard econometric methods assume stable relationships. Recent ML research offers tools designed for exactly this setting: conformal prediction methods that provide calibrated uncertainty intervals even under distribution shift (Barber et al., 2023; recent work at NeurIPS 2025 on conformal prediction with change points), distributional robustness methods that stress-test predictions against worst-case subgroup shifts (Sagawa et al., 2020), and continual learning frameworks that update models as new data arrives (Parisi et al., 2019). We find these methods compelling precisely because they address a limitation that the existing economics tools were not designed to handle: prediction when the regime is changing.
>
> (c) Capability benchmarking: The ML community is already building economically grounded benchmarks. SWE-bench measures AI on realistic coding workflows. TheAgentCompany (NeurIPS 2025) and SWE-Lancer (2025) evaluate agents on real professional tasks with actual economic value. These demonstrate that capability trajectories on work-relevant tasks can be tracked and evaluated. Such prediction and benchmarking could be expanded to broader occupation-work flow contexts and even organizations.
>
> 2. Limitations/Ethical Consideration: The topic of AI and labor is increasingly becoming a politicized topic, and sensational claims that are not peer-reviewed can have outsized influence (e.g., the recent Citrini Research report causing dramatic drop in the stock prices of SAS companies). As such, predicting AI’s impact on the labor market needs to be done in a rigorous and transparent manner. The scientific community will need to ensure that the research is evidence-based, rigorously conducted, and peer-reviewed, before sharing research findings with the public. The parallel to climate science is informative. Scientists anchored public understanding by building transparent, reproducible prediction infrastructure around climate change.
>
> 3. Interdisciplinary collaboration: First, platforms: dedicated tracks and workshops at leading venues like ICML would signal that this is a legitimate research direction and create space for cross-disciplinary exchange. Second, funding: currently, grants for studying AI's labor impact are largely directed at economists using traditional methods (e.g., RCTs). Funding agencies are not explicitly calling for ML-economics collaboration. Demonstrating that such collaboration produces results, through the kinds of shared tasks and benchmarks we propose, is the most effective way to shift funding incentives. Third, there is a natural cascade: if ICML embraces this direction, other disciplines, industry, and funders will follow, because the ML community is at the frontier of AI development and its signals carry weight across fields.
>
> To sum up our planned revisions are: (1) Concrete examples subsection in Section 4 covering agent-based simulation, prediction under non-stationarity, and capability benchmarking. (2) Expanded discussion of limitations and ethical considerations. (3) Add discussion of more concrete interdisciplinary mechanisms.

---

> > ### Author Rebuttal · Reviewer_942X · 2026-04-03
> >
> > Thank the authors for answering my questions and addressing my concerns. I’m happy with the responses.

---

### Decision · Program_Chairs · 2026-04-30

**Decision:**

Accept (regular)

**Comment:**

The decision is to accept the paper.

The paper advocates for treating prediction of AI's societal effects as a core machine learning problem, rather than externalizing it to other fields such as economics. The central argument is that the ML field has many methodological, cultural, and institutional resources to offer, as well as insider knowledge of the technology itself, and that these can fill current gaps in the way that such forecasting might be done external to the ML community.

After the rebuttal stage, reviewers reached consensus that this paper stakes out a position that it would be valuable to discuss, and could drive interest into an important and productive area. The authors made several commitments to sharpen, elaborate, and clarify some arguments, and I strongly encourage them to follow through.